# Structural Connectivity Reorganization Based on DTI after Cingulotomy in Obsessive–Compulsive Disorder

**DOI:** 10.3390/brainsci13010044

**Published:** 2022-12-24

**Authors:** Sara Kierońska-Siwak, Paweł Sokal, Magdalena Jabłońska, Marcin Rudaś, Agnieszka Bylinka

**Affiliations:** 1Department of Neurosurgery and Neurology, Collegium Medicum, Nicolaus Copernicus University, 85-168 Bydgoszcz, Poland; 2Department of Neurosurgery and Neurology, Jan Biziel University Hospital No 2, Collegium Medicum, Nicolaus Copernicus University, 85-168 Bydgoszcz, Poland; 3Doctoral School of Medical and Health Sciences, Collegium Medicum, Nicolaus Copernicus University, 85-067 Bydgoszcz, Poland; 4Department of Radiology and Imaging Diagnostics, Jan Biziel University Hospital No. 2, 85-168 Bydgoszcz, Poland

**Keywords:** diffusion-tensor imaging, tractography, arcuate fasciculus, cingulotomy, forceps minor, obsessive–compulsive disorder, transcranial direct current stimulation

## Abstract

Bilateral cingulotomy is a procedure applied to patients with obsessive–compulsive disorder (OCD). This report presents the structural changes occurring within the forceps minor and arcuate fascicles nerve fibers after a successful bilateral anterior cingulotomy in the patient with refractory OCD. Cingulotomy mainly affects the values of FA, MD, and ADC in the treatment of the examined nerve bundles. This structural reorganization coexists with a good clinical effect. However, it is necessary to expand the study group and to investigate the correlation between the parameters of diffusion and anisotropy and the patient’s clinical condition (Y-BOCS scale).

## 1. Introduction

Obsessive–compulsive disorder (OCD) is a chronic mental illness with widespread prevalence in the general population. OCD as a disease entity was first described in 1838 by Jean-Étienne Dominique Esquirol. It is estimated that the prevalence of OCD in the world population ranges from 1.5% to 3% [1,2]. The mean age of onset is 19.5 years. In adolescence, more cases of OCD are reported in men, while among adults, women suffer more frequently [3]. Moreover, there the literature shows that women are at higher risk of developing OCD throughout their lives than men [4,5,6]. 

Patients are diagnosed according to the guidelines specified in the Diagnostic and Statistical Manual of Mental Disorders-Fifth Edition (DSM-V), while the Yale–Brown Obsessive-Compulsive Scale (Y-BOCS) is used to determine the severity of symptoms [6,7]. The basic therapeutic process combines two complementary components: pharmacological and psychotherapeutic. Psychopharmacological treatment mainly involves the use of a serotonin reuptake inhibitor (SRI) and selective serotonin reuptake inhibitors (SSRI), while psychotherapeutic intervention is based on the recognized cognitive–behavioral therapy in the form of Exposure and Response Prevention (ERP) [3,8,9]. 

It is estimated that despite relatively effective conservative therapy, up to 20% of patients are resistant to it [10]. Thus, patients with treatment-resistant form of OCD constitute a special group of patients who require looking for other therapeutic solutions. The alternative and approved treatment strategies include techniques of deep brain stimulation, i.e., repetitive transcranial magnetic stimulation (rTMS) and transcranial direct current stimulation (tDCS), as well as methods in the field of psychoneurosurgery, cingulotomy, or capsulotomy [10,11,12,13,14].

Anterior bilateral cingulotomy is one of the methods of psychoneurosurgery used in the treatment of OCD resistant to the standard treatment strategy. This technique involves stereotaxic, bilateral, and thermal (using a thermoelectrode) damage to nerve fibers in the middle part of the cingulate gyrus, just above the roof of the lateral ventricles. Anterior bilateral cingulotomy is used in the treatment of OCD refractory to the standard treatment strategy [15]. The target of neuroablation is determined using magnetic resonance imaging and is usually 7 mm from the midline of the cingulate gyrus and 20–25 mm posterior to the apex of the frontal horn [16].

Diffusion tensor imaging (DTI) is an imaging technique that enables the non-invasive visualization of the course of nerve fibers. The basis of its operation is detecting Brownian motion. These are diffusive movements of water molecules in the extracellular space [17,18,19]. The degree of orientation of diffusion of water molecules is called fractional anisotropy (FA). Within the white matter of the CNS, the FA parameter reaches high values, which reflects its degree of order. MD is the parameter to describe the amount of diffusion in each measured voxel. The value of this parameter depends on the tissue density and not on the spatial orientation of the fibers [17,18,19].

The purpose of this case report is to present the microstructural changes occurring within the forceps minor and arcuate fascicles nerve fibers after a successful bilateral anterior cingulotomy in a patient with refractory OCD.

## 2. Materials and Methods

### 2.1. Case History

The discussed case concerns a 46-year-old man diagnosed in 2011 with obsessive–compulsive disorder. The patient was admitted to the Department of Neurosurgery at University Hospital No. 2 in Bydgoszcz to perform an anterior cingulotomy. The patient reported that the first symptoms of the disease appeared around the age of 20. In the initial stage, the disturbances mainly involved obsessions related to the need to maintain cleanliness of the environment and cleaning the space in which he stayed. At around 30 years of age, the patient’s symptoms progressed. The severity of the disorders was manifested by the necessity to wash hands frequently and to bathe the whole body several times a day. The final diagnosis—obsessive–compulsive disorder–was made during the patient’s first hospitalization in a psychiatric ward, based on the DSM-V criteria. The patient was then 35 years old. From the moment of diagnosis, the patient was under psychiatric and psychotherapeutic care. The pharmacological treatment initially consisted of fluoxetine at a dose of 20 mg for 16 weeks, after which no improvement was noted. The treatment was extended to include clomipramine at a dose of 40 mg for 16 weeks, but again no improvement was observed. It was decided to add risperidone at a dose of 20 mg to the therapy, which consequently was associated with a slight improvement in reducing obsessions and compulsions. In addition, the patient attended cognitive–behavioral therapy (CBT) for 48 weeks throughout his treatment.

Upon admission to the clinic, the patient reported that he was washing his hands approximately 50 times a day. In addition, the patient reported obsessions with the fear of bacteria.

The physical examination showed abrasions of the epidermis and wounds on the skin of the entire hands. Moreover, during admission to the ward, the patient was assessed on the Y-BOCS Obsession and Compulsion scale and on the Global Assessment of Functioning scale (GAFs), in which he obtained 33 and 40 points, respectively.

### 2.2. MRI and DTI Acquisition

The patient was imaged at 3.0 T (Philips Ingenia, Aera scanner, Erlangen, Germany) using a 32 channel head coil. Head was scanned without any angulation ( in every direction: AP, RL, FH angle was 0°). 

The axial DTI sequence was performed with the following parameters: scan type: imaging; scan mode: MS; scan technique: SE; acquisition mode: cartesian; fast imaging mode: EPI; EPI factor: 45; shot mode: single-shot; diffusion mode: DTI; gradient coil: no; directional resolution: medium (15); number of b- factors: 2: b1-0 b2-800 Echoes: 1; TE/TR: shortest: 85/3232 (ms); slice thickness: 2.5 (mm); slice gap: 0 (mm); number of signal averages (NSA): 2; phase encoding: AP; FOV: 224 (FH) × 224 mm (AP) × 140 mm (RL); acquisition matrix: 92 × 90; reconstruction matrix: 128; acquisition voxel size: 2.43 mm (RL), 2.49 mm (AP), 2.50 mm (FH); recon voxel size: 1.75 mm (RL), 1.75 mm (AP), 2.50 mm (FH).

### 2.3. Fiber Tracking Protocol

Diffusion tensor images were processed using DSI studio software, BSD License ((http://dsi-studio.labsolver.org, accessed on 28 November 2022). A DTI diffusion scheme was used and a total of 60 diffusion sampling directions were acquired. Longer tracts. The b-value was 1000 s/mm^2^. The in-plane resolution was 1.95 mm. The slice thickness was 2 mm. A deterministic fiber tracking was used. A total of 15,000 tracts were calculated. ROIs were defined automatically based on an anatomical atlas loaded into the DSI studio program. By determining forceps minor, the first Region of Interest (ROI) was drawn in genu of corpus callosum.

### 2.4. Cingulotomy Procedure

The patient underwent stereotactic bilateral anterior cingulotomy using stereotactic frame and software with automatic CT/MRI image fusion and RF Neuro Generator (Cosman Medical, Burlington, MA, USA 2015) for electrocoagulation.

Targets for the electrode tip were positioned bilaterally by using MRI guidance (Inversion Recovery and T1-weighted contrast enhanced sequences), typically 7 mm lateral to the midline, 20–25 mm posterior to the frontal horn of the lateral ventricle, just above the roof of the ventricle.

The electrode was implanted through a burr hole located approximately 2 cm to midline anterior to the coronal suture.

The electrode settled in one target point was heated to 80 Celsius degrees for 90 s. The lesion was enlarged by coagulation in next three points every 2 millimeters above target along electrode trajectory.

Thereafter symmetrical lesion was created in opposite hemisphere in the same way. 

A planning of cingulotomy procedure is presented in Figure 1.

Outcome after treatment and after follow up.

### 2.5. Limitation

In the described article, the limitation is mainly the short time of patient observation—one step follow up. This is mainly due to the distance of the patient’s residence from the hospital. In the future, our goal is to expand the study group and increase follow-up with controls after 6, 12, and 24 months after cingulotomy. In addition, our next goal is to broaden the assessment of the patient’s functioning and the severity of symptoms by conducting more frequent psychological consultations.

The limitations associated with cingulotomy also result from the need to develop better cooperation between psychiatrists who treat patients with OCD pharmacologically and neurosurgeons performing the cingulotomy procedure.

Referral of patients with OCD from psychiatric outpatient clinics to neurosurgical departments performing surgical procedures is still a problem.

## 3. Results

In the presented case, the analysis covered forceps minor and arcuate fasciculus. The following parameters were used for the purpose of the evaluation: FA, ADC, MD, and the number of fibers. The parameters were measured immediately before the procedure and six months after the cingulotomy. Their exact values are presented in Table 1 and Table 2.

Based on the collected results, after 6 months follow up, an increase in the FA value of both the forceps minor and both the arcuate fasciculus was noted. Moreover, an increase in the number of fibers was observed in each analyzed structure, compared to the initial results.

Changes in the anatomy of forceps minor before and after cingulotomy are presented in Figure 2 and Figure 3; moreover, arcuate fasciculus before and after cingulotomy is presented in Figure 4 and Figure 5.

In addition, we observed a decrease in the apparent diffusion coefficient—the ADC parameter, in both examined structures compared to the value before neuroablation. We also noted that there was a reduction in the MD parameter after cingulotomy.

Moreover, there were significant changes reported by the patient himself during the interview. According to the subjective assessment of the patient, the obsession with dirt and bacteria decreased. As a consequence, the patient noticed a reduction in compulsive behaviors, which resulted in a nearly 40% reduction in water consumption costs. Another promising step made by the patient after the surgery was an attempt to re-activate professionally and efforts to increase social activity. The presented progress was also associated with a reduction in the doses of drugs taken. 

Postoperative clinical evaluation also included reassessment of the patient according to the following scales: YBOCS, GAF, and HDRS. Six months after a successful cingulotomy, the patient’s YBOCS score dropped by 18 points compared to the baseline result. The decrease in scores was also reported according to the HDRS scale. On the other hand, in the Global Assessment of Functioning scale, we noted an increase in the score by 21 points. The improvement of the obtained results corresponded to the improvement of the clinical condition manifested by the patient.

The patient’s qualification for cingulotomy is carried out with the participation of a psychiatrist and a neurologist.

We perform cingulotomy in the case of insufficient control of the symptoms of the disease in a patient who has previously been treated with SSRI drugs, neuroleptics in the correct doses, and cognitive–behavioral therapy for a minimum of 1 year.

Common transient adverse effects described in many publications during cingulotomy included urinary incontinence and confusion/disorientation, subsiding within days postoperatively. Serious adverse complications included seizure in less than 5%, hemiparesis in less than 1%, and personality changes in less than 1% of operations reported across all studies [20].

Observed other side effects included headaches, nausea, vomiting, and seizures. While a spectrum of cognitive side effects has been reported following cingulotomy, cases with no adverse effects have also been documented [21].

Proper planning of the target in the cingulotomy procedure and the accuracy of its implementation is a way to avoid these side effects. It is also important to properly hydrate the patient before and after the procedure and to maintain normal blood pressure to avoid reduced cerebral perfusion.

## 4. Discussion

To the best of our knowledge, this is the first publication to assess morphological changes in white matter structures, including forceps minor and arcuate fascicles, using DTI. The available data from the literature indicate a significant advantage of research worker interest over the clinical implications of cingulotomy, while much less attention is paid to reorganization as a result of neuroablative surgery. Based on the above arguments, we are convinced that this paper will shed new light on the potential relationship between the reorganization of white matter structures resulting from successful bilateral cingulotomy and the presented clinical improvement in patients with drug-resistant obsessive–compulsive disorder.

This report presents the microstructural changes occurring within the forceps minor and arcuate fascicles nerve fibers after a successful bilateral anterior cingulotomy in the patient with OCD refractory to pharmacotherapy. Over the past three decades, significant progress has been made in understanding the neurobiological underpinnings of OCD. Neuroimaging techniques have proven to be helpful tools [22]. The results of meta-analyses and systematic reviews, both on functional MRI and those using diffusion tensor imaging (DTI), indicate that the dominant—but not the only area involved in the pathogenesis of OCD—is the fronto-striato-thalamo-cortical circuitry [23,24]. However, as the available research supporting DTI imaging shows, other areas of the brain are also involved in the development of OCD. Focal changes were found in the areas of the temporal, parietal and occipital cortex, dorsolateral/medial prefrontal cortex, and in the internal capsule and the anterior part of the insula [25]. A meta-analysis of 30 studies conducted by Zhang et al. showed a significant decrease in FA in the right hemisphere in the gyrus rectus and lenticular nucleus, while in the left hemisphere of the brain in the superior frontal gyrus (orbital part) and superior longitudinal fasciculus. Moreover, a decreased value of FA was demonstrated in the corpus callosum and in the right cerebellar hemispheric lobule [26]. These data indicate that the rich symptomology of OCD is likely to be reflected in the abnormalities of extensive areas of the brain. The available data, although abundant, often show discrepancies in results, possibly reflecting differences in the methods adopted by the researchers for conducting studies with the use of DTI. Nevertheless, most of the available results indicate a decrease in the FA parameter, especially in the cingulate bundle, the corpus callosum, and the anterior limb of the internal capsule [27,28,29,30]. A decrease in the FA parameter value may suggest a decrease in the integrity of the white matter forming the above-mentioned structures. Moreover, the results of the research by Versace et al. showed a reduced value of the FA cingulum bundle compared to healthy participants. However, this parameter was not decreased along the entire length of the bundle, but only the focal length in its middle part. In addition, decreased cingulum bundle collinearity was noted in both hemispheres of the brain. This research also demonstrated an increased projection of the cingulum bundle fibers into the prefrontal cortex. These results may suggest that the loss of concentricity of the cingulum bundle fibers is reflected in the decrease in FA of the described area [31].

The meta-analysis conducted by Radu et al. also showed abnormalities in the fibers building the cingulum bundle and corpus callosum, but also in SLF, middle and inferior longitudinal fasciculi, frontal aslant tract and/or inferior fronto-occipital fasciculus, anterior thalamic radiation, and internal capsule. Particular attention was paid to the reduced values of the FA parameter within the intersection of cingulum bundle and corpus callosum. Moreover, these changes were more pronounced among patients taking drugs from the group of selective serotonin reuptake inhibitors (SSRIs) [32]. Similar data were obtained in the study by Benedetti et al. These data suggest the role of drugs in modulating the microstructure of white matter fibers through their influence on the differentiation and remyelination of oligodendrocytes [33]. These processes would result in a decrease in the value of the FA parameter while increasing the value of the MD parameter [32,33].

A study by He et al. with the use of TRACULA (TRActs Constrained by UnderLying Anatomy) in the evaluation of the results of probabilistic treatment, showed that in patients with untreated OCD, the FA forceps minor parameter was lower compared to healthy participants in the study. The lowest FA values measured in the forceps minor were recorded in patients presenting with severe OCD. It is also important that the value of the FA forceps minor parameter was inversely correlated with the degree of OCD severity. Moreover, this analysis showed significantly reduced values of FA and AD parameters right uncinate fasciculus in untreated participants, indicating its participation in the pathogenesis of OCD [34].

The presented case of the patient is innovative due to the comparison of changes in tractography before and after cingulotomy.

In presented case we recorded an increase in the value of FA both in forceps minor and arcuate fascicles after 6 months after cingulotomy what coexist with clinical improvement. High FA values are found in normal white matter fibers. They reflect the directed diffusion of water molecules along the white matter fibers. A decrease in this parameter suggests the presence of white matter damage or disorganization. In addition, we noted an increased value of MD. Based on the available literature data, these changes may be the result of overlapping changes in the microstructure of forceps minor white matter fibers and therapy with SSRI drugs.

## 5. Conclusions

Structural connectivity changes after a successful bilateral cingulotomy in the patient with refractory OCD occur within the forceps minor and arcuate fascicles nerve fibers. Cingulotomy is a procedure that mainly affects the values of FA, MD, and ADC in the treatment of the examined nerve bundles, which coexists with a good clinical effect. However, it is necessary to expand the study group and to investigate the correlation between the parameters of diffusion and anisotropy and the patient’s clinical condition (Y-BOCS scale).

## Figures and Tables

**Figure 1 brainsci-13-00044-f001:**
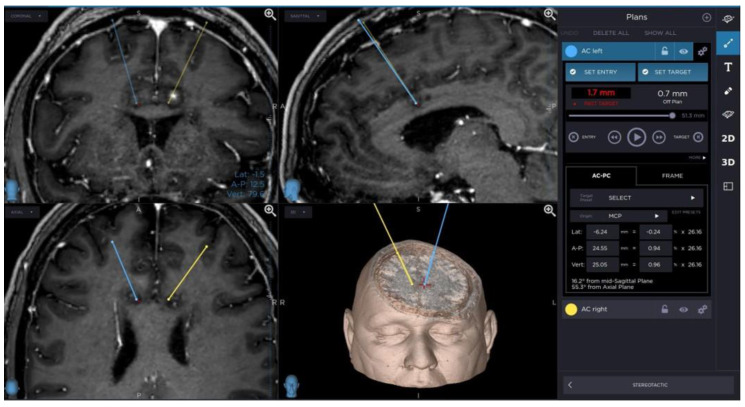
Planning of cingulotomy procedure.

**Figure 2 brainsci-13-00044-f002:**
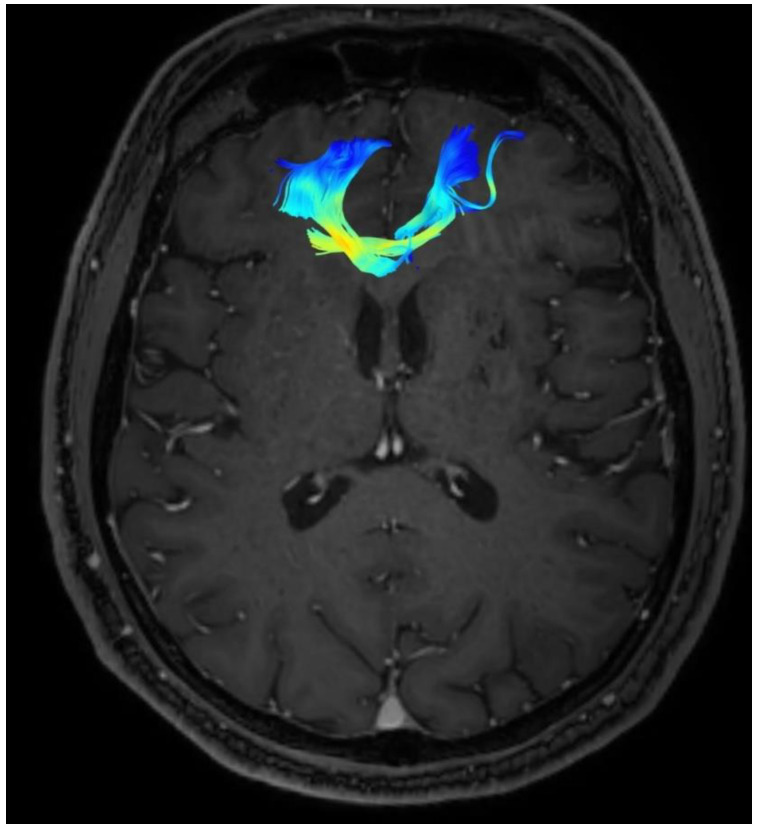
Forceps minor before cingulotomy.

**Figure 3 brainsci-13-00044-f003:**
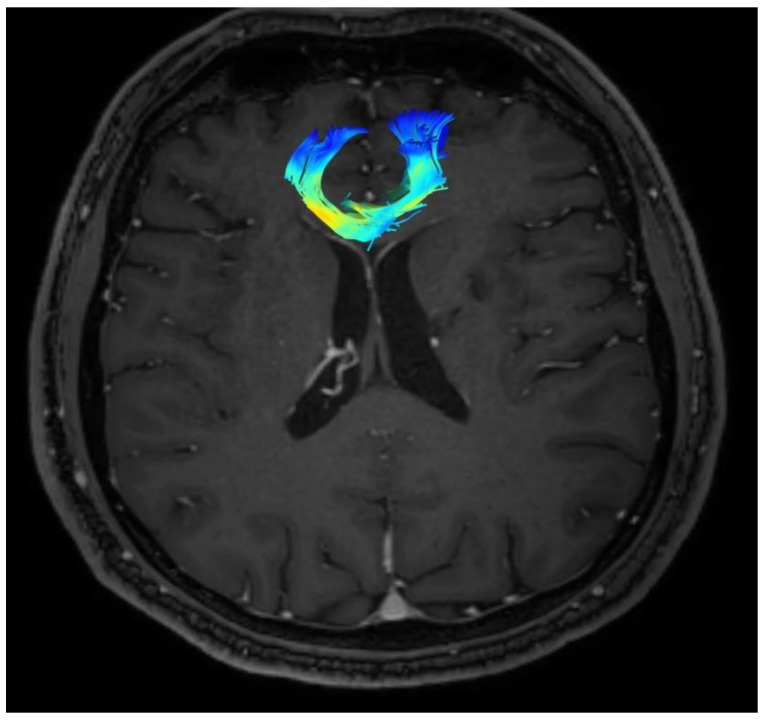
Forceps minor after cingulotomy.

**Figure 4 brainsci-13-00044-f004:**
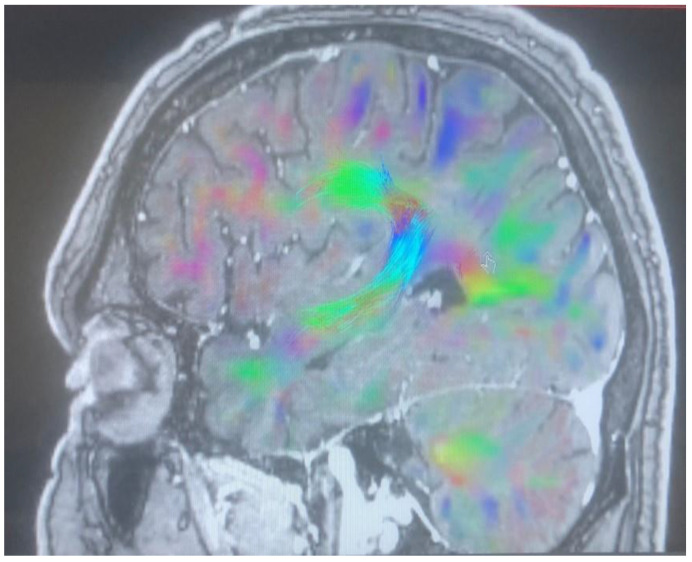
Arcuate fasciculus before cingulotomy.

**Figure 5 brainsci-13-00044-f005:**
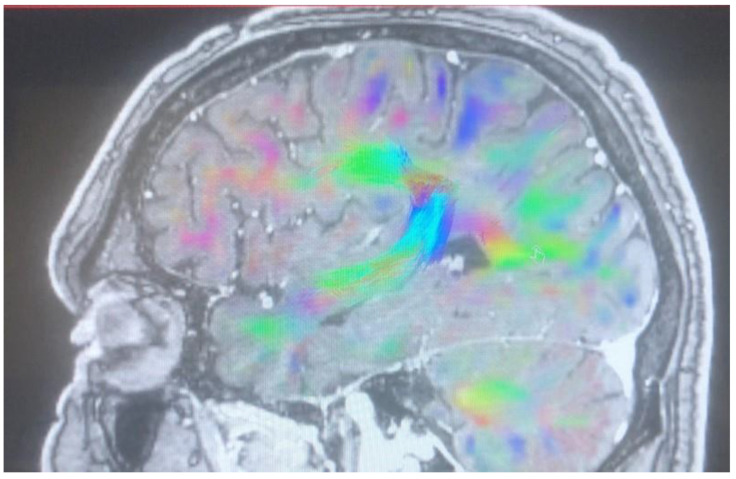
Arcuate fasciculus after cingulotomy.

**Table 1 brainsci-13-00044-t001:** The patient’s morphological data of the forceps minor before cingulotomy and 6 months follow up.

Parameters	Before Cingulotomy	6 Months Follow Up
FA	0.62	0.77
MD	0.825	0.798
ADC	0.735	0.622
Number of fibers	176	183

FA—fractional anisotropy, MD—mean diffusivity, ADC—apparent diffusion coefficient

**Table 2 brainsci-13-00044-t002:** The patient’s morphological data of the arcuate fascicles before cingulotomy and 6 months follow up.

Parameters	Before Cingulotomy		6 Months Follow Up	
Side	Left	Right	Left	Right
FA	0.53	0.65	0.67	0.75
MD	0.807	0.815	0.780	0.795
ADC	0.810	0.805	0.742	0.752
Number of fibers	202	205	210	213

## Data Availability

Not applicable.

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
