# Peer review of "Structural Connectivity Reorganization Based on DTI after Cingulotomy in Obsessive–Compulsive Disorder"

_brainsci, 2022, doi:10.3390/brainsci13010044_

Round 1
Reviewer 1 Report
The authors reported the microstructural changes occurring within the forceps minor and arcuate fascicles nerve fibers after a successful bilateral anterior cingulotomy in a patient with obsessive-compulsive disorder (OCD) refractory to pharmacotherapy. This is an interesting and promising manuscript which gives amazing points of comprehensive review instantly to the readers about the topic. I have the following suggestions for this manuscript.
l Why should this case be reported? Please confirm if your case adds new knowledge to the literature. The authors need to provide compelling reasons in the manuscript why their case merits publication. I would appreciate to see the creation of a new table to summarize the literature review to better describe why this case report is unique and be reported. Does this paper add any new information to our surgical knowledge? The authors must restructure the paper and also emphasize why this clinical case is essential and what brings really new.
l The authors have mentioned their operative measures in treating refractory OCD. Please discuss in depth the indications for surgery and how to select or combine different measures when treating the disease.
l Please discuss the potential danger of performing your procedure and how to avoid it.
l Some of the references are outdated and should be updated accordingly.
l I would appreciate to see the insertion of an abbreviation list.
Author Response
Thank you very much for the professional review and valuable comments on the article. As suggested by the reviewer, the following changes were made:
- To the best of our knowledge, this is the first publication to assess morphological changes in white matter structures, including forceps minor and arcuate fascicles, using DTI. The available literature data indicate a significant advantage of research worker interest over the clinical implications of cingulotomy, while much less attention is paid to reorganization as a result of neuroablative surgery. Based on the above arguments, we are convinced that this paper will shed new light on the potential relationship between the reorganization of white matter structures resulting from successful bilateral cingulotomy and the presented clinical improvement in patients with drug-resistant obsessive-compulsive disorder
- We add paragraph regarding the patient's qualification for cingulotomy:
“The patient's qualification for cingulotomy is carried out with the participation of a psychiatrist and a neurologist.
We perform cingulotomy in the case of insufficient control of the symptoms of the disease in a patient who has previously been treated with SSRI drugs, neuroleptics in the correct doses and cognitive-behavioral therapy for a minimum of 1 year.”
- We discussed also potential danger of the procedure
“Common transient adverse effects described in many publications during cingulotomy included urinary incontinence and confusion/disorientation, subsiding within days postoperatively. Serious complications adverse included seizure in less than 5%, hemiparesis in less than 1%, and personality change in less than 1% of operations reported across all studies.
Observed other side effects include headaches, nausea, vomiting, and seizures. While a spectrum of cognitive side effects have been reported following cingulotomy, cases with no adverse effects have been documented.”
- The references has been updated
We hope that the introduced changes will contribute to a significant improvement in the quality of the article and will make it eligible for publication.
- We add Abbreviation list
We hope that the introduced changes will contribute to a significant improvement in the quality of the article and will make it eligible for publication
Reviewer 2 Report
Dear Editor,
I really appreciate the opportunity to review the manuscript brainsci-2097891 entitled:
"Structural connectivity reorganization based on DTI after cingulotomy in obsessive-compulsive disorder"
I commend the authors for describing this critical and timely issue. The paper is interesting and well-written; however, I would like to highlight some issues that merit revision:
Nowhere are the limitations indicated; please the authors to add a short statement where they are indicated; among these should be added the short period of post-surgical follow-up, and only one step of follow-up. It would be interesting, and for this I urge the authors, in addition to the present study, which is very good for all the remaining parts, to prepare a second paper after a 12-, possibly 24-month follow-up, so as to see any psychopathological settling in the patient after the impact with everyday life for a longer period.
Overall a very good paper.
Author Response
Thank you very much for the professional review and valuable comments on the article. As suggested by the reviewer, the following changes were made:
- Paragraph with limitations has been added as suggested:
Limitation
In the described article, the limitation is mainly the short time of patient observation - one step follow up. This is mainly due to the distance of the patient's residence from the hospital. In the future, our goal is to expand the study group and increase follow-up with controls after 6, 12 and 24 months after cingulotomy. In addition, our next goal is to broaden the assessment of the patient's functioning and the severity of symptoms by conducting more frequent psychological consultations.
The limitations associated with cingulotomy also result from the need to develop better cooperation between psychiatrists who treat patients with ocd pharmacologically and neurosurgeons performing the cingulotomy procedure.
Referral of patients with ocd from psychiatric outpatient clinics to neurosurgical departments performing surgical procedures is still a problem.
We hope that the introduced changes will contribute to a significant improvement in the quality of the article and will make it eligible for publication.